



# Ocean circulation, sea ice, and productivity simulated in Jones Sound, Canadian Arctic Archipelago, between 2003-2016

Tyler Pelle[1], Paul G. Myers[2], Andrew Hamilton[2], Matthew Mazloff[1], Krista M. Soderlund[3], Lucas Beem[3], Donald D. Blankenship[3], Cyril Grima[3], Feras Habbal[3], Mark Skidmore[4], and Jamin S. Greenbaum[1]

[1]Scripps Institution of Oceanography, University of California, San Diego, La Jolla, CA, USA
[2]Department of Earth and Atmospheric Sciences, University of Alberta, Edmonton, Canada
[3]Institute for Geophysics, Jackson School of Geosciences, University of Texas at Austin, Austin, TX, USA
[4]Department of Earth Sciences, Montana State University, Bozeman, Montana, USA

**Correspondence:** Tyler Pelle (tpelle@ucsd.edu)

**Abstract.** Jones Sound is one of three critical waterways that regulate liquid exchange between the Arctic and northern Atlantic Oceans within the Canadian Arctic Archipelago. However, to date, no high-resolution ocean circulation model exists to study the recent evolution of Jones Sound, meaning that our understanding of circulation within the sound is based either on temporally and spatially sparse oceanographic observations or on extrapolating conditions within Baffin Bay, which has a more dense observational record. To address this, we developed a high-resolution ($1/120°$, 0.9 km) Jones Sound configuration of the Massachusetts Institute of Technology general circulation model and performed coupled ocean-sea ice-biological productivity simulations between 2003-2016 to investigate recent changes within this waterway. We find that circulation through Lady Ann Strait, Fram Sound, and Glacier Strait comprise 75%, 14%, and 11% of the volumetric transport into and out of Jones Sound, with tidal flushing enhancing the magnitude and temporal variability of volumetric transport through all three waterways. Warming Atlantic Water within western Baffin Bay flows into Jones Sound through Lady Ann Strait, becomes well-mixed, and circulates counter-clockwise, encroaching on the terminus of most tidewater glaciers that line the eastern periphery of the sound. Furthermore, we find that sustained atmospheric and oceanic warming drive an 11% reduction in the summertime sea ice extent, decreased wintertime sea ice thickness, and delayed onset of sea ice refreeze in the fall (thus lengthening the amount of time in which Jones Sound is ice free). Tidal flushing through Cardigan Strait is critical in triggering meltback of sea ice across northern Jones Sound. Lastly, this decline in sea ice increases light availability and coupled with warming of the subsurface waters in Jones Sound, facilitates enhanced primary productivity at ocean levels down to ~21 meters depth. While we note that the modeled warming signal in Baffin Bay is overestimated compared to observations, the results presented here improve our general understanding of how this critical waterway might change under continued polar amplified global warming and underscores the need for sustained oceanographic observations in this region.





## 1 Introduction

The Canadian Arctic Archipelago (CAA) is a tangle of shallow basins and narrow straits that connect the Arctic Ocean to Baffin Bay and the northern Atlantic Ocean. The Queen Elizabeth Islands (QEI) in the north is an area with relatively small islands surrounded by the larger Ellesmere, Devon, Cornwallis, Bathurst, Melville and Prince Patrick islands. The waters of the Arctic Ocean flow through the QEI and are transported into northern Baffin Bay (and eventually the north Atlantic Ocean)

through three main passageways: Lancaster Sound, Nares Strait, and Jones Sound. Within these waterways, moorings reveal a mean transport of 0.46 Sv (1 Sv is equal to $10^6$ m$^3$/s) in western Lancaster Sound between 1998-2010 (Peterson et al., 2012; Prinsenberg and Hamilton, 2005), $0.71 \pm 0.09$ and $1.03 \pm 0.11$ Sv between 2003 to 2006 and 2007 to 2009, respectively, along Nares Strait (Munchow and Humfrey, 2008; Münchow, 2016), and 0.3 Sv between 1998-2002 flowing through Jones Sound. Thus, the rough balance of these three main passages is 46% Nares Strait, 34% Lancaster Sound and 20% through Jones Sound

(Melling et al., 2008; Grivault et al., 2018).

Jones Sound is home to the Inuit hamlet of Ausuiktuq (Grise Fiord) and is a marine region surrounded by glaciers draining large ice fields and ice caps on both the Ellesmere and Devon Islands. It is the third largest export pathway in the CAA (Grivault et al., 2018; Melling et al., 2008; Zhang et al., 2016), connecting directly to the Arctic Ocean at the narrow (15 km) and shallow (150 m) western gateways of Hellsgate/Cardigan Strait (which merge into Fram Sound) and exchanging with Baffin Bay on

the east side of the sound at a depth of 450 m (figure 1). The western half of Jones Sound is shallow, being ∼200 m depth, while the eastern basin is deeper, having a maximum depth of ∼840 m and greater exchange with external waterways. Aside from facilitating liquid exchange between the Arctic and northern Atlantic Oceans, Jones Sound also hosts a diverse biological ecosystem that is sustained in part by ice-ocean interactions of tidewater glacier termini as well as seasonally-varying oceanic and sea ice conditions (Bhatia et al., 2021). However, as part of the broader Arctic, Jones Sound is vulnerable to changing

climate conditions that threaten these natural resources. For instance, Gardner et al. (2012) found that ice mass loss across the QEI has tripled from $31 \pm 8$ Gt/y in 2004-2006 to $92 \pm 12$ Gt/y in 2007-2009, largely driven by Arctic amplified atmospheric warming that outpaces the global average by three-times. Below the halocline within the Baffin Bay, mid-depth Atlantic Water (AW) that penetrates into the CAA and Arctic Ocean has been warming steadily since at least the early 2000s (Polyakov et al., 2013; Wang et al., 2020; Ballinger et al., 2022). This atmospheric and oceanic warming has also driven recent declines in sea

ice extent, with the summer sea ice area in northern Canadian waters and Baffin Bay declining at a rate of 7.1% per decade and 11.6% per decade, respectively (Tivy et al., 2011). Given that sea ice serves as hunting platforms for polar bears, resting grounds and nursery areas for walruses and seals, and hosts for algae that grow on ice base that are important to the marine food supply, these reductions in sea ice have cascading impacts on marine ecosystems.

Due to the maze of islands, narrow straits, complex coastlines, and shallow bathymetry, modeling ocean circulation, sea

ice dynamics, biological productivity, and their joint interactions within the CAA is challenging but remains one of the best ways to begin understanding recent change of CAA waterways. Recent ocean modeling studies are largely capable of resolving mean transport through Lancaster Sound and Nares Strait (McGeehan and Maslowski, 2012; Shroyer et al., 2015; Wang et al., 2017; Wekerle et al., 2013; Zhang et al., 2016); however, coarse horizontal meshes cause issues with resolving baroclinic flow



Figure map

**Figure 1.** Domain featuring ocean bathymetry (m) from the SRTM15+ dataset (Tozer et al., 2019), where blue shading denotes bathymetry below sea level. Gray shading denotes the present-day extent of land ice, the red box denotes the domain of our ocean circulation model, the black box denotes where new bathymetry observations were implemented in the ocean model in Sverdrup Bay. Geographic features mentioned in the text are labeled, including glaciers, water ways, and islands.

in narrower channels, such as Fram Sound, leading to large uncertainty in volume estimates into and general circulation within

Jones Sound. This, in turn, limits the fidelity of sea ice and productivity models in this region. To address this, we develop a high-resolution, Jones Sound configuration of the Massachusetts Institute of Technology general circulation model and perform coupled ocean-sea ice-biological productivity simulations between 2003-2016 to study the following: (1) The magnitude and spatial/temporal distribution of volumetric transport into and out of Jones Sound; (2) Fine-scale circulation features within Jones Sound and their impact on watercolumn structure on seasonal timescales; (3) Seasonal and decadal variations in sea

ice dynamics; (4) The impact of simulated changes in ocean and sea ice conditions on productivity within Jones Sound. By investigating these four topics, we seek to improve our understanding of circulation, sea ice dynamics, and productivity within



Jones Sound and how it fits within the broader context of the Canadian Arctic Archipelago. Below, we provide information on the set up of the numerical ocean, sea ice, and biogeochemical model as well as an overview of the simulations.

(1) How have external atmospheric and far-field oceanic forcing conditions driven change of ocean circulation and sea ice dynamics within Jones Sound between 2003-2016? (2) What are the magnitude and spatial/temporal distributions of volumetric (3) Have simulated changes in ocean and sea ice conditions driven change in Jones Sound oceanic productivity through 2016?

## 2   Methods

Here, we provide information on the setup of the ocean, sea ice, and productivity model that we use to simulate circulation in Jones Sound between 2003-2016. We note that we first developed a low resolution ocean-sea ice model to simulate the region surrounding Jones Sound between 2002-2016 (hereon referred to as the "low resolution simulation"), from which we extract initial and boundary conditions to force our high-resolution ocean-sea ice-productivity simulation of Jones Sound from 2003-2016 (hereon referred to as the "high-resolution simulation"). Most of the model setup between the two simulations is identical aside from the source of the boundary and initial conditions as well as the domain extent and resolution.

### 2.1   Ocean model setup

We model ocean circulation using a regional Canadian Arctic configuration of the ocean component of the Massachusetts Institute of Technology general circulation model (MITgcm; Marshall et al., 1997). This includes use of the hydrostatic approximation, a dynamic/thermodynamic model to simulate sea ice dynamics (Losch et al., 2010), and the Biogeochemistry with Light, Iron, Nutrients, and Gases with Nitrogen (N-BLING) productivity module to simulate photosynthetic biological productivity (only used in the high-resolution simulation and described in section 2.2; Galbraith et al., 2010). The low resolution simulation domain extends from 73.25 - 76.75°S, 74 - 91.5°E, has a nominal horizontal resolution of 1/24 degree (∼5 km) and contains 70 vertical levels (with vertical resolution of 7 m through 266 m depth, then decreasing to a minimum resolution of 62 m at the lowest ocean level; 1090 m). The high-resolution simulation domain extends from 75.40 – 76.70°S, -77.45 – -91.45°E (red outline in figure 1), has a nominal horizontal resolution of 1/120 degree (∼900 m) and contains 70 vertical levels (with vertical resolution of 7 m through 266 m depth, then decreasing to a minimum resolution of 56 m at the lowest depth of 963 m). Model bathymetry is based on the SRTM15+ digital elevation model (Tozer et al., 2019) with corrections applied in Brae Bay (black box near Sverdrup Glacier in figure 1) and the oceanic regions near Grise Fiord following multibeam and point measurement (mostly conductivity, temperature, and depth sensor casts) bathymetric observations. Ocean and sea ice parameter values that differ from Nakayama et al. (2018) are provided in table 1.

Initial and boundary conditions for the low resolution ocean simulation are extracted from the 1/12 degree Arctic and Northern Hemisphere Atlantic (ANHA12) configuration of the NEMO model that covers the entirety of the Northern Atlantic Ocean down to 20°S and was run from 2002-2018 (Hu et al., 2019; Gillard et al., 2020). Fields used as initial and boundary conditions in our ocean simulations include temperature, salinity, the zonal (u) and meridional (v) velocity components, and sea surface height. In addition, fields used to initialize and drive the sea ice model include sea ice area, thickness, snow content, salt



**Table 1.** MITgcm ocean and sea ice model parameters and values used in this study. Only parameters that are different from Nakayama et al. (2018) are shown.

| Parameter (unit) | Value |
| --- | --- |
| Ocean/air drag coefficient scaling factor | 0.00125 |
| Air/sea ice drag coefficient | 0.00125 |
| Lead closing (m) | 1 |
| Sea ice dry albedo | 0.72 |
| Sea ice wet albedo | 0.63 |
| Snow dry albedo | 0.78 |
| Snow wet albedo | 0.65 |
| Ocean emissivity | 0.97 |
| Ice emissivity | 0.95 |
| Snow emissivity | 0.95 |

content, and the u and v sea ice velocity components. Initial conditions were extracted at model date Jan. 1, 2002 and boundary
conditions are extracted monthly through Dec. 31, 2016. Initial and bi-monthly boundary conditions in the high-resolution
simulation are extracted from the low resolution simulation, with an initialization date of Jan. 1, 2003.

Atmospheric forcing is taken in three-hour intervals from the Arctic System Reanalysis version 2 (ASRv2; Bromwich et al.,
2018) and interpolated onto the model grids. We use the following variables: 2 m air temperature, 2 m specific humidity,
precipitation, 10 m u and v wind components, short/long wave radiation, atmospheric pressure, evaporation, and river/glacial
runoff (time series of atmospheric forcing shown in Figure 2). In addition, tidal forcing (amplitude and phase) is prescribed in
the low resolution simulation using the Arctic 2 kilometer Tide Model (Arc2kmTM; Howard and Padman, 2021) and includes
the following constituents: $M_2$, $S_2$, $N_2$, $K_2$, $K_1$, $O_1$, $P_1$, and $Q_1$. In MITgcm, tidal forcing is applied along the ocean model
boundaries as a surface elevation perturbation and as barotropic flow that propagates throughout the domain.

## 2.2 N-BLING setup

The Nitrogen version of the Biogeochemistry with Light, Iron, Nutrients, and Gases (N-BLING) productivity module simulates
biogeochemical cycling of key elements/micronutrients as well as photosynthetic productivity and has been implemented in
MITgcm as a module (Galbraith et al., 2010; Verdy and Mazloff, 2017). N-BLING is driven by the physical ocean model
as well as atmospheric carbon dioxide concentrations, which are taken monthly between 2003-2016 from a meteorological
station in Alert, Canada and assumed to be spatially uniform across our high-resolution domain. Incoming solar radiation, taken
from ASRv2, also drives N-BLING. Initial and monthly boundary conditions for N-BLING include the following: dissolved
inorganic carbon (DIC) and alkalinity taken from the Global Ocean Data Analysis Project version 2 (GLODAPv2; Olsen et al.,
2016); $O_2$, $NO_3$, $PO_4$, and Silica taken from the World Ocean Atlas (WOA; Garcia et al., 2018); Fe, dissolved organic nitrogen,



dissolved organic phosphorus, and initial small/large/diazotroph phytoplankton from a global run of BLINGv2 (Pers. Comm. Eric Galbraith); and iron dust deposition from Mahowald et al. (2005). N-BLING runs on the same computational grid and timestep as the ocean model and coupling is only one-way, meaning that N-BLING-simulated productivity does not influence the radiative fluxes, and thus the ocean circulation and sea ice growth.

### 2.3 Model runs

We first run the low resolution ocean-sea ice simulation between Jan. 1, 2002 to Dec. 31, 2016 using a baroclinic timestep of 120 seconds. We performed simulations that do and do not resolve tidal forcing to explore the impact of tides on ocean circulation and sea ice productivity in Jones Sound (see Appendix-a). From the low resolution tidal simulation, we extract initial and boundary conditions to force the high-resolution ocean-sea ice-biological productivity simulation, which is run between January 1, 2003 to Dec. 31, 2016 using a baroclinic timestep of 70 seconds. All results discussed below are taken from the high-resolution simulation. We note that N-BLING crashed in May 2015, so we only report productivity results up to then.

## 3 Results

### 3.1 Model Forcing

Between 2002-2016, there exist large trends in atmospheric forcing variables in ASRv2 over Jones Sound that are applied in our simulations. The time series of all atmospheric forcing variables are shown in figure 2, where domain-wide means are taken except for evaporation, precipitation, and runoff, which are summed. In panels a-i, we overlay the overall and summer linear trends as colored solid and dotted lines, respectively, and report the total trend in the title of each panel. For panel-j, we normalize each time series between 0 and 1 and recompute the total and summer trends of each variable. Lastly, we shift each line so that they start at 0 to facilitate comparison between the trends. Overall, we find strong increasing trends in the 10 m air temperature, which is increasing at an overall rate of 0.87°C/decade and a summer rate of 1.19°C/decade (1.37 times higher than the overall trend). This strong surface warming trend drives strong positive trends in summer relative humidity (3.76e-4 kg/kg/decade; 2.81 times stronger than the overall relative humidity trend), summer and overall evaporation, and summer runoff (6.583e-4 m/s/decade). We see limited changes in mean atmospheric pressure, mean short and long wave radiation, mean 2 m wind speed, and total integrated precipitation.

Along the ocean model boundaries, we find large changes in the ocean state between 2002 and 2016 as simulated by the ANHA12 configuration of NEMO, in which we take our oceanic boundary conditions. Figure A1 shows vertical profiles of ocean potential temperature applied to the Eastern boundary (-74°W) of the low resolution ocean simulation in Jan. of 2002, 2007, 2012, and 2016. A strong warming signal of the winter mid-depth Atlantic Water (between 100-300 m depth) is evident after 2007, where waters warm by over 2°C in ∼10 years in this simulation. We note that this simulated warming signal of the Atlantic Water is overestimated, as oceanic observations taken on the Arcticnet Amundsen Ice Breaker report 1-1.5°C of



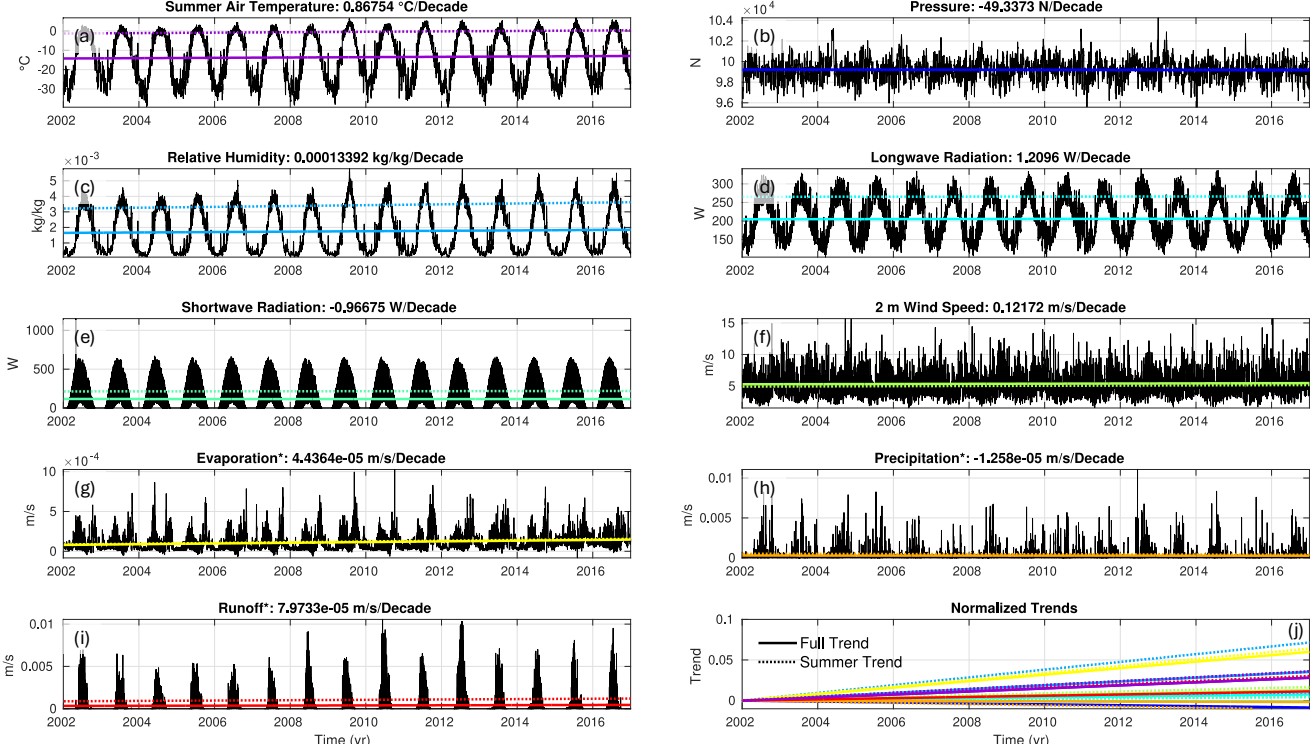

**Figure 2.** Time series of ASRv2 (a) 10 m air temperature, (b) surface atmospheric pressure, (c) relative humidity, (d) downward longwave radiation, (e) downward shortwave radiation, (f) 2 m wind speed, (g) evaporation, (h) precipitation and (i) runoff. Panels a-f show domain-wide averages while panels g-i (names appended with "*") show sums integrated across the domain. In each panel, the best fit lines for all data (solid colored line) and for summer data (dotted colored line) are overlaid and the slope of the complete line (per decade) is included in the title. Panel-j plots all best-fit lines computed on data that have been normalized between 0 and 1. The lines are then vertically shifted so that the initial value is 0 to facilitate trend comparison.

warming of this water mass in northern Baffin Bay through this same time period. We further discuss the implications of this overestimated ocean warming in the discussion section.

## 3.2 Model Evaluation

We evaluate our coupled ocean-sea ice model of Jones Sound against repeat Summertime casts of Conductivity, Temperature, Depth (CTD) ocean sensors made aboard the Amundsen Ice Breaker between 2005 and 2021 (Amu, 2003-2021). We selected four sample sites based on data availability within our model domain: two locations spanning longitudinally across northern Baffin Bay, one site in the center of the eastern entrance of Lancaster Sound, and one site near the center of Jones Sound. We plot these observed (solid lines) temperature-depth profiles in figure 3a-d together with the modeled profiles (dotted lines) taken close to the same date to facilitate comparison between the modeled and observed ocean state. The color of the profile





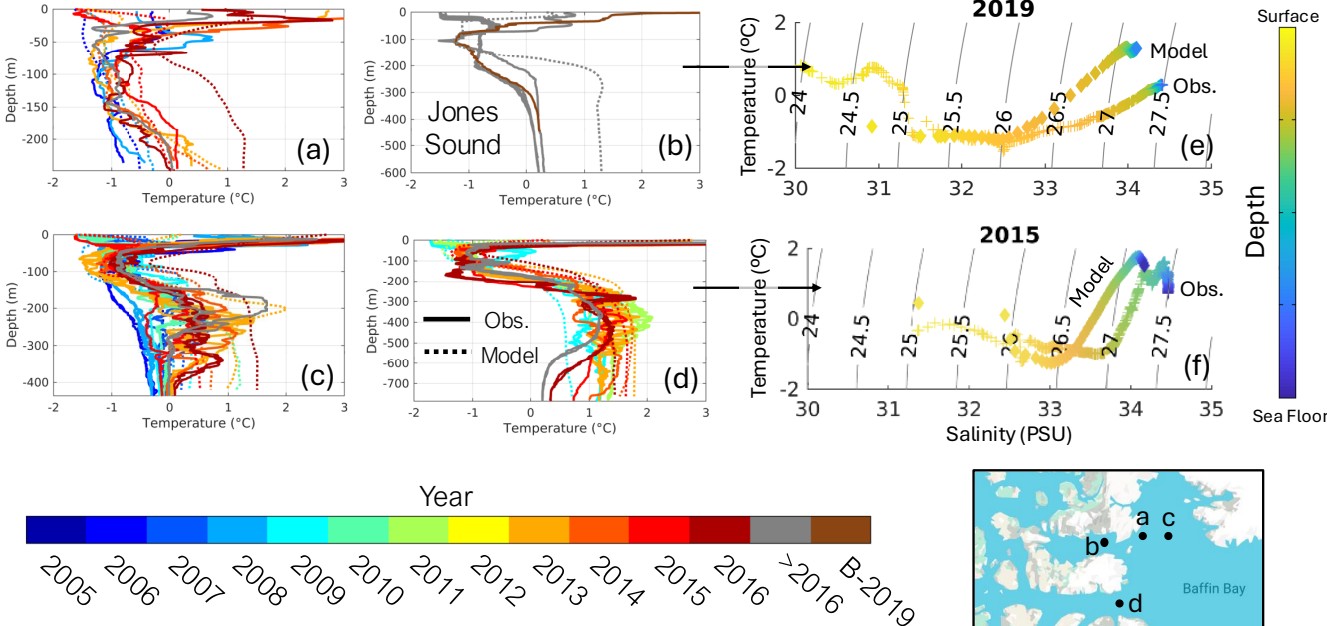

**Figure 3.** Comparison of observed and modeled (solid and dotted lines, respectively) ocean temperature vertical profiles taken at (a/c) two sample locations in northern Baffin Bay, (b) Jones Sound, and (c) the eastern entrance of Lancaster Sound. The color of the line indicates the year of the profile and gray denotes any profile taken after 2016 (the dotted gray lines are the summer 2016 modeled profile). Observations are summertime CTD-rosettes taken aboard the Amundsen Ice Breaker (Amu, 2003-2021) and the brown line in panel-b is a profile from Bhatia et al. (2021) that were taken in summer 2019 (B-2019). (e-f) Temperature-salinity plots taken in (e) 2019 in Jones Sound and (f) 2015 in the eastern entrance of Lancaster Sound. The color of the marker corresponds to the depth and observed versus modeled profiles are labeled.

is associated with the year it was taken. Note that gray solid profiles indicate observed profiles taken after 2016 and dotted gray profiles were extracted in summer 2016. We included these profiles because the only currently publicly available CTD casts within Jones Sound in this dataset were taken in 2019 and 2021. Brown lines correspond to profiles taken near Sverdrup Glacier and Belcher Glacier in the summer of 2019 by Bhatia et al. (2021). Note that we also present evaluation of salinity profiles in Appendix A1 (figure A2).

In the profiles taken across northern Baffin Bay and in Lancaster Sound (figure 3a/c/d), we find that our model is generally able to replicate the summertime thermal properties of the ocean in these sample locations. In particular, we properly model the depth of the thermocline across all sample sites (located between 100-300 m depth) and model temperatures that are within the summer observational range between 2005-2016. One notable exception to this is a warm bias in the modeled seabed ocean temperatures in the sample site in Northern Baffin Bay (figure 3c), where observed ocean temperatures increase from -0.5°C to -1°C between 2005-2016 but modeled ocean temperatures increase from 0°C to 1.5°C during this same time period. That is, we overestimate warming of the bottom ocean water in Baffin Bay but reasonably match warming of the mid-depth Atlantic



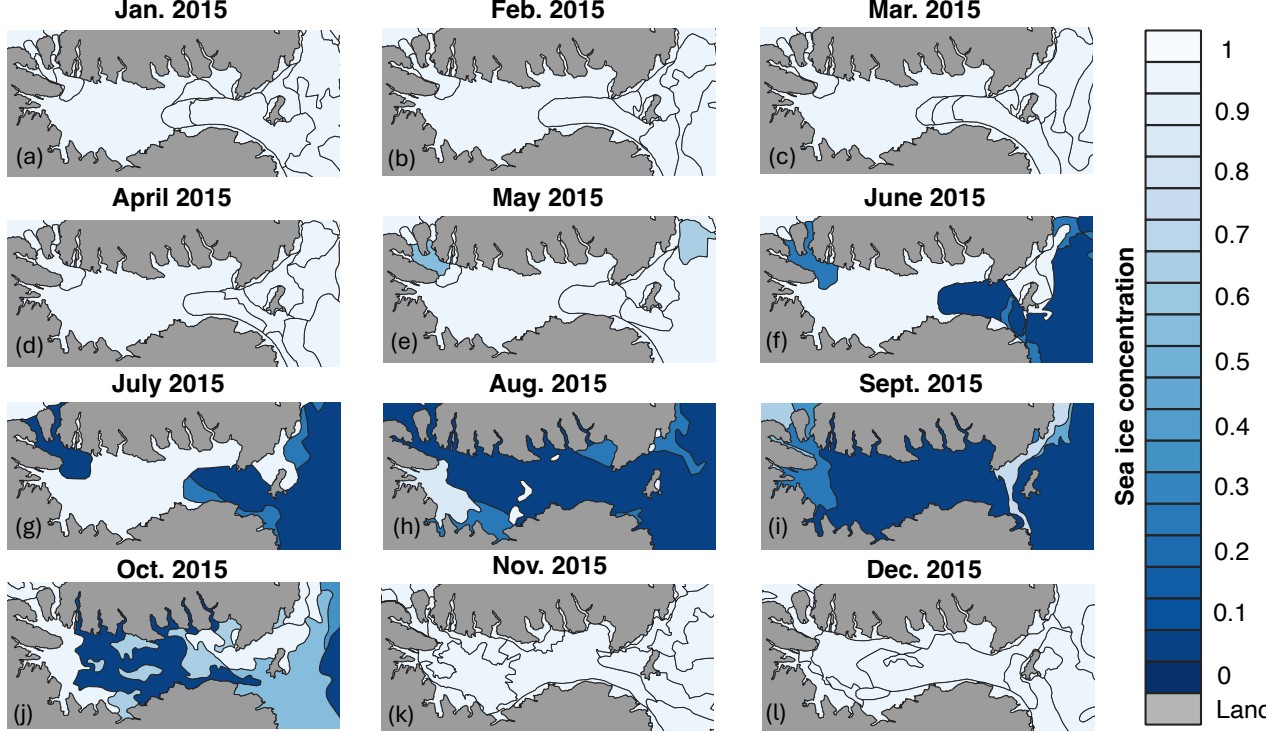

**Figure 4.** Monthly snapshots of sea ice concentration observations in 2015 from the Canadian Ice Service, which are created through the manual analysis of in situ, satellite, and aerial reconnaissance data (Service, 2009). Gray shading denotes land.

water in the summer. We expect that this overestimated bottom ocean warming is a result of the overemphasized wintertime warming of the mid-depth Atlantic Water (figure A1) in our model boundary conditions, as the ANHA NEMO ocean model run also has this bias. Within Jones Sound, observations are limited to profiles taken in 2019 and 2021 as well as 2006, 2007, and 2019, respectively. However, from the available data, we find that this warm bias of Baffin Bay water masses in our model also extends into these waterways. In both cases, while we correctly model the depth of the thermocline, we find that ocean

temperatures below the thermocline are ∼1°C too warm at the end of our simulation (figure A4a/b). Furthermore, we note that simulated vertical profiles of salinity generally show good agreement with observed profiles (figure A2 and figure 3e-f). However, we note that modeled salinity gradients with depth are weaker than observed. For instance, in Lancaster Sound, ocean waters below ∼300 m are too fresh by ∼0.5 PSU. These reduced salinity gradients could impact the magnitude of current velocities in our model. However, as we correctly simulate the depth of the thermocline, thermal properties of water

above the thermocline, and salinity gradients in mixed-layer, we believe the modeled circulation, the extent to which warmer waters can spatially extend, and surface properties of the ocean model are realistic.

     We further evaluate the sea ice model by comparing monthly observed (figure 4) and modeled (figure 5) timestamps of 2015 sea ice concentration. While we only show the 2015 sea ice cycle, we note that the other yearly cycles feature similar spatial



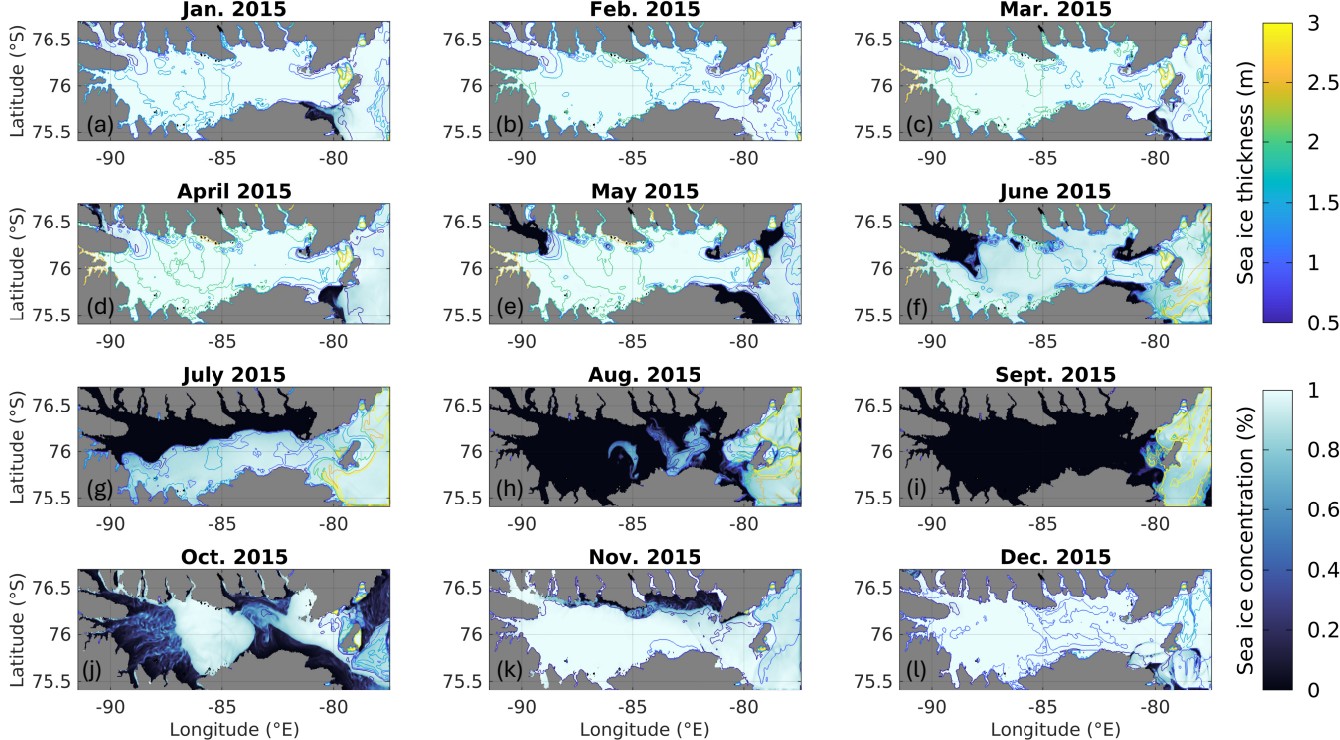

**Figure 5.** Monthly timestamps of simulated surface ocean grid cell sea ice concentration and contoured sea ice thickness (contoured every 0.5 m). Gray shading indicates land.

and temporal patterns of annual change. Observed 2015 sea ice concentration fields are taken from the Canadian Ice Service
and are created through manual analysis of in situ, satellite, and aerial reconnaissance data (Service, 2009). We find generally
good agreement between the observed and modeled sea ice concentration states within Jones Sound for the 2015 sea ice cycle,
with complete sea ice coverage simulated across the domain between November-April and near-complete meltback of sea ice
in August and September. Importantly, we also find that our model is able to replicate the spatial pattern of sea ice meltback,
with flow through Fram Sound initiating sea ice decline in northwestern Jones Sound and partial clearing of sea ice across
South-Eastern Jones Sound. We note that modeled sea ice thickness east of Coburg Island is overestimated and sea ice does not
clear out of this region during the summer months, disagreeing with observations. As such, in section 3.4, we only integrate
sea ice quantities west of Coburg Island.

## 3.3 Ocean circulation through Jones Sound

Circulation into and out of Jones Sound is dictated by flow through three waterways: Fram Sound (the confluence of Cardigan
Strait and Hell Gate) in the northwest, Lady Ann Strait through the southeast, and Glacier Strait through the northeast. In
figure 6a-d, we show timeseries (a-b) and temporal-mean bar plots (c-d) of volumetric transport integrated through these three





gateways into Jones Sound (lines in figure 8a). Of the temporal-mean 3.54 Sv that flows into and out of Jones Sound, flow through Lady Ann Strait, Fram Sound, and Glacier Strait comprise 75.14% (2.66 Sv), 13.84% (0.49 Sv), and 11.02% (0.39 Sv) of this volumetric transport, respectively figure 6). Flow through each of these waterways is a mix of inflow and outflow into

the sound; Lady Ann Strait is comprised of $\sim 1.16 \pm 1.17$ Sv inflow and $\sim 1.50 \pm 1.27$ Sv outflow, Fram Sound is comprised of $\sim 0.24 \pm 0.27$ Sv inflow and $\sim 0.25 \pm 0.24$ Sv outflow, and Glacier Strait is comprised of $\sim 0.27 \pm 0.24$ Sv inflow and $\sim 0.12 \pm 0.14$ Sv outflow. Of particular importance is the influence of tidal flushing on transport through these waterways, enhancing the total transport through Lady Ann Strait, Fram Sound, and Glacier Strait by 57%, 308%, and 63%, respectively. In comparing figure 6a-b, we see that tidal forcing drives large temporal-variation in net transport through all three waterways.

In particular, tidal flushing through Lady Ann Strait drives inflow transport magnitudes as high as $\sim$5 Sv near the start of 2014 and outflow magnitudes as high as $sim$4.5 Sv. When tidal fluctuations are removed, we find that both Fram Sound and Glacier Strait comprise mainly inflow into Jones Sound, while Lady Ann Strait comprises 40% inflow and 60% outflow (figure 6d).

In figure 6e-f, we plot vertical cross sections of the 2003-2016 mean velocity fields through transects across Fram Sound (e), Lady Ann Strait (f), and Glacier Strait (g; yellow, red, and green lines in figure 8a, respectively) to visualize regions of flow

into (red, positive) and out of (blue, negative) Jones Sound. Time-mean ocean velocities through Fram Sound are the highest across all three waterways (maximum velocity of $\sim$0.7 m/s), with regions of inflow occupying the top 65 m of the water column and outflow located below (figure 6e). That is, while Fram Sound is the smallest waterway leading to Jones Sound by cross-sectional area, it has the second greatest integrated volumetric transport passing through it because of these high flow velocities. For Lady Ann Strait, the primary region of inflow is located near southern Coburg Island (the northern end of Lady

Ann Strait) and persists through depth (figure 6f). We model a secondary weak region of inflow on the southern end of the strait below 200 m depth. Otherwise, outflow dominates circulation through the strait, with the strongest outflow comprised primarily by a persistent strong current that wraps around eastern Devon Island. Flow through Glacier Strait is dominated by inflow into Jones Sound, with a small region of outflow modeled on the extreme southern-end of this waterway (figure 6g). Overall, these results highlight the complex spatial and temporal regimes of transport into and out of Jones Sound.

In figure 7a-f, we show March (a-c) and September (d-f) mean ocean velocity fields that are averaged through 150 m depth at year 2014 (a/d), 2015 (b/e), and 2016 (c/f). We selected March and September to capture circulation through Jones Sound with complete and no sea ice coverage, respectively. In March, we see similar circulation patterns through 150 m in Jones Sound, with strong inflow ($sim$0.35 m/s) Fram Sound in the west and strong outflow (>0.5 m/s) through Lady Ann Strait in the east. Flow of strong currents from northern Baffin Bay tend to flow to the east of Coburg Island rather than through Glacier Strait,

although we do flow of 0.20 m/s in March 2016 (figure 7). Within the sound under full sea ice coverage, the flow field is rather quiescent. In September, however, reduced sea ice coverage allows atmosphere-ocean interactions, which drives the formation of persistent eddies that dominate circulation within Jones Sound (figure 7d-f). These eddies do not always form in the same place and rotate in the same direction. For instance, in 2014 (figure 7d), we model three sustained cyclonic eddies (rotating counterclockwise), where the easternmost eddy interacts with strong inflow through Glacier Strait. In 2015 (figure 7e), we

observe one weak cyclonic eddy in eastern Jones Sound and note that circulation through Glacier Strait flows around Cohburg Island and connects to strong outflow through Lady Ann Strait. Lastly, in 2016 (figure 7f), we model three anticyclonic (rotating







**Figure 6.** Time series of net volumetric transport through Fram Sound (blue), Glacier Strait (green), and Lady Ann Strait (red) with (a) tidal forcing included and (b) tidal forcing not included. Positive and negative values indicate transport into and out of Jones Sound and note the different y-axis limits between the two panels. (c-d) Bar plots of the temporal-mean transport into (blue) out of (orange) Jones Sound, with the mean net transport indicated by the yellow bar. Panel-c includes tides and panel-d does not include tides. Error bars insicate one standard deviation from the mean. (e-f) Vertical cross sections of time-mean ocean velocity through (e) Fram Sound, (f) Lady Ann Strait, and (g) Glacier Strait.

clockwise) eddies and one cyclonic eddy that interacts with strong inflow through Glacier Strait. The diversity of summertime circulation patterns across 2014-2016 in Jones Sound highlights the complex dependence of current-patterns on inflow/outflow regimes and ocean-atmosphere interactions.

These circulation patterns influence the thermodynamic structure of water-masses within Jones Sound. In figure 7g-l, we plot vertical temperature profiles across Jones Sound at 76°N (red line in figure 7a). As in the velocity fields, the third and fourth row



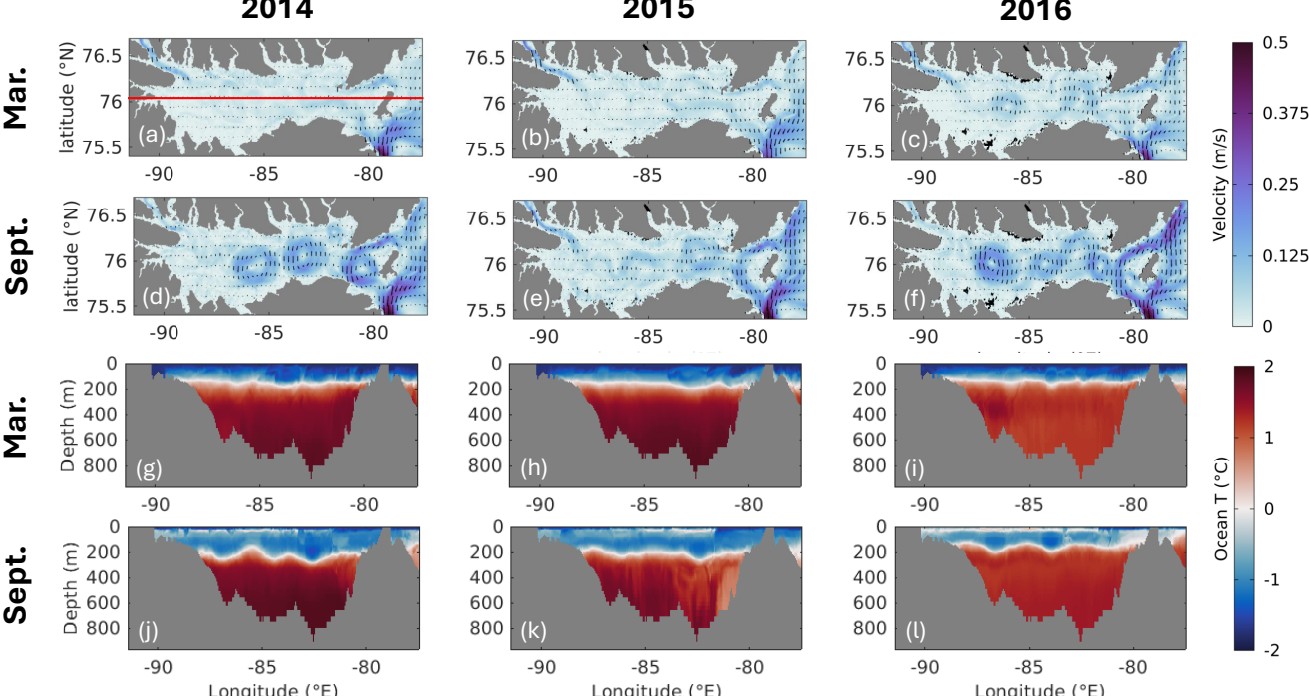

**Figure 7.** (a-f) Time and depth (down to 150 m) mean ocean velocity (m/s) fields with flow vectors overlaid. Panels a-c and d-f show March and September mean velocity fields, respectively. (g-l) Vertical profiles of ocean temperature (°C) taken along the red line in panel-a. Note that the first, second, and third columns show 2014, 2015, and 2016 mean fields.

of the figure correspond to March and September mean temperature fields and the first, second, and third columns correspond to 2014, 2015, and 2016 fields. In all temperature fields, we note that the modeled thermal structure is homogeneous beneath the thermocline, as has been observed in Jones Sound (figure 3f). Under complete sea ice coverage, the modeled thermocline is nearly uniform at a depth of ∼100 m across Jones Sound (figure 7g-i). In September, the development of eddies drives spatial variation in the modeled depth of the thermocline, with the greatest variation simulated in 2014 and 2016 (figure 7j/l). In 2015, when the September velocity field is closely matched to the March 2016 field, we note that the thermocline is nearly uniform at ∼200 m depth across Jones Sound, ∼100 m deeper than during March.

The simulated spatial reach of warm waters below the thermocline in Jones Sound is topographically constrained. In figure 8, we plot the modeled ocean bottom temperature and velocity vectors in March and September of 2010, 2013, and 2016. As noted in the validation above, our model overestimates warming of waters below the thermocline in Jones Sound; however, as we correctly model the depth of the thermocline, we intend for the temperature fields shown here to be viewed as tracers of "warm" ocean water and its general circulation throughout the sound. It is evident that warm ocean waters flow into the model domain from the eastern model boundary, highlighting that the source of warm water inflow into Jones Sound is via northern





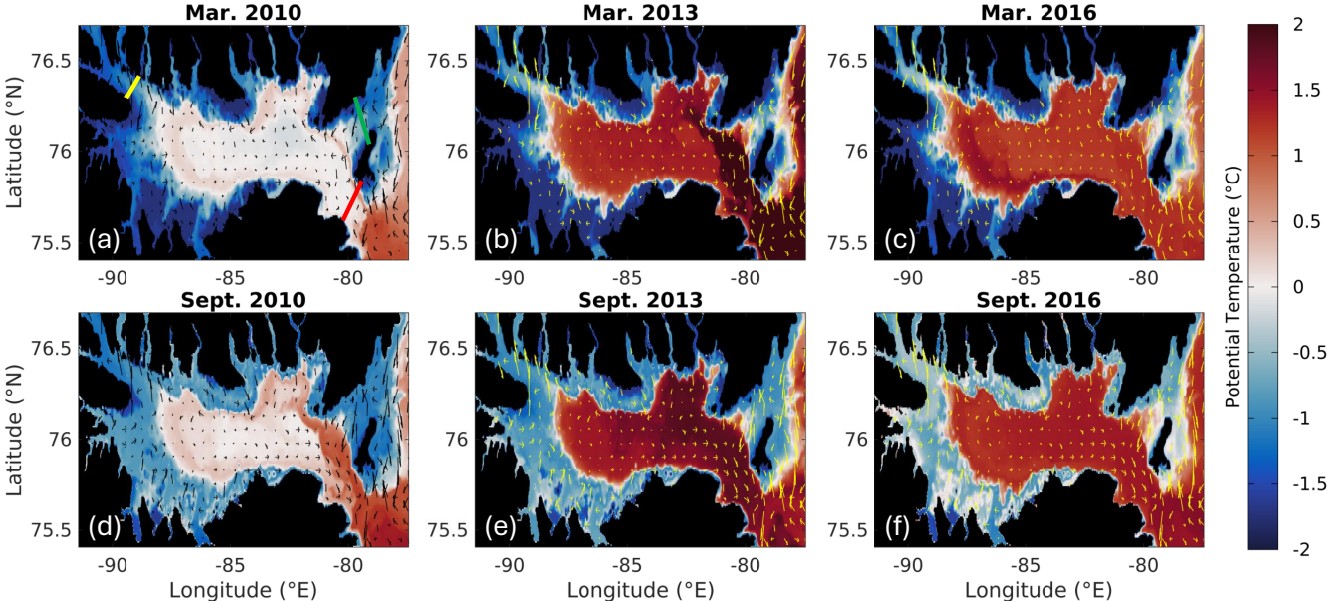

**Figure 8.** Modeled ocean bottom potential temperature (shaded) and velocity (vectors) in Jones Sound, Canadian Arctic Archipelago. The top row shows fields in March of (a) 2010, (b) 2013, and (c) 2016 while the bottom row shows fields in September (d) 2010, (e) 2013, and (f) 2016. In panel-a, the yellow, red, and green lines denote the location of the vertical cross sections through Fram Sound, Lady Ann Strait, and Glacier Strait that are shown in figure 6e-g, respectively.

Baffin Bay. This warm water then circulates into Jones Sound from the southeast through the deep bathymetry (>600 m depth; figure 1) that underlies Lady Ann Strait. Inflow of deep warm ocean water is blocked by shallow bathymetry along Glacier Strait, which sits between 100-300 m depth. Once the warm bottom water intrudes into Jones Sound after 2010, it circulates counter-clockwise and although topographically constrained, warm bottom water breaches the entrance of most major fiords along the Eastern periphery of the Sound by 2012 (where bathymetry is generally deeper than that of the western sound).

Specifically, warm bottom water reaches the terminii of Sverdrup, Jakeman, and Belcher Glaciers (figure A3). Depression of the thermocline in September relative to March constrains the reach of this warm bottom water. This can be seen in Brae Bay and western Jones Sound when comparing figure 8b and figure 8e. In regions where this warm water cannot circulate (regions of bathymetry above the thermocline), waters are ~0.5°C warmer due to enhanced atmospheric mixing.

## 3.4   Sea ice decline

Sea ice is a prominent feature of ocean circulation and dynamics within Jones Sound, regulating the exchange of heat, moisture, and momentum between the ocean and atmosphere, as well as blocking the penetration of sunlight that fuels photosynthetic biological productivity. In figure 5, we plot monthly timestamps of grid cell sea ice concentration and contoured sea ice thickness during model year 2015. Sea ice undergoes a yearly cycle in which the sound becomes largely ice free in September





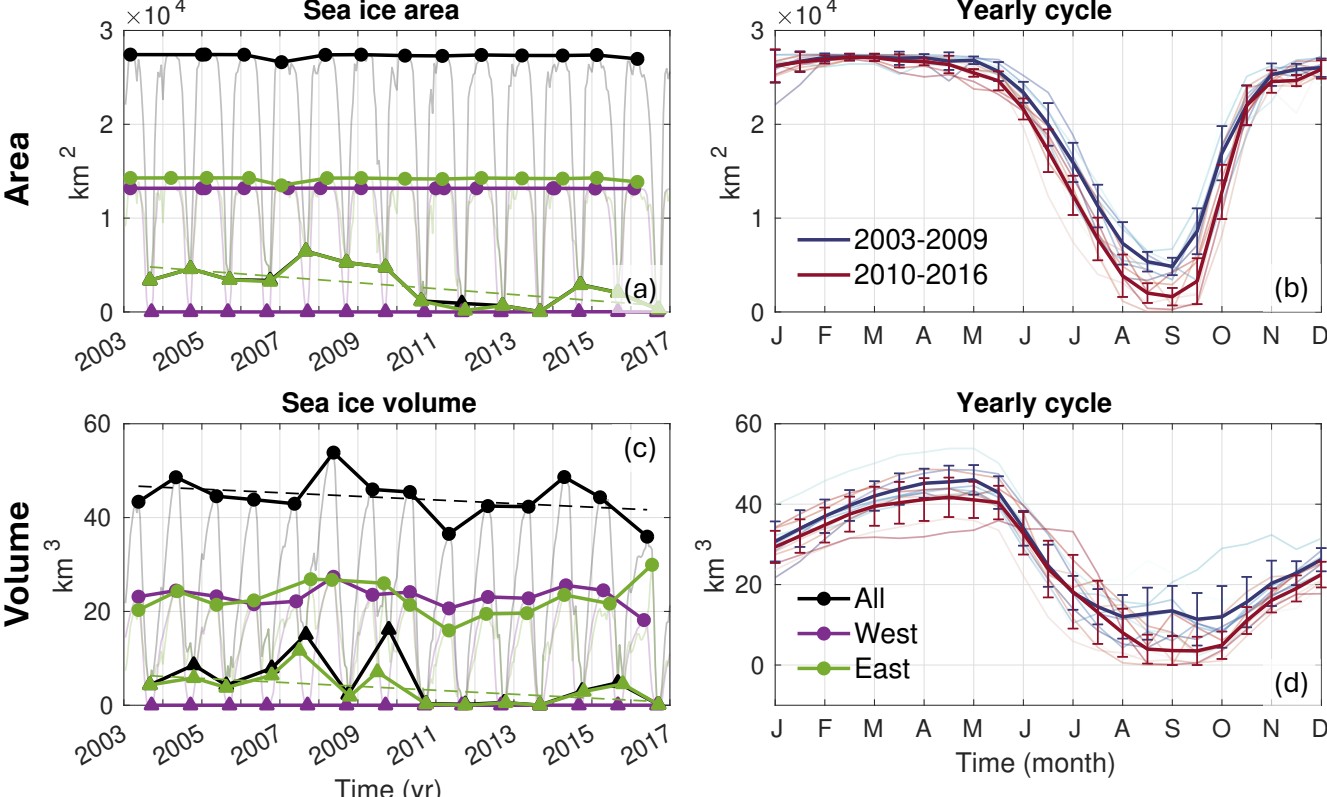

**Figure 9.** Integrated sea ice (a-b) area and (c-d) extent. In panels a/c, marker colors correspond to the area of integration: black (all of Jones Sound), purple (western Jones Sound), and green (eastern Jones Sound). Circles and triangles denote yearly sea ice maxima and minima. Best fit lines are overlaid on the East Jones Sound area and volume minima (green dashed lines) and the full Jones Sound volume maxima (black dashed line) to show long-term decreases. In panels b/d, the 2003-2009 and 2010-2016 mean yearly cycles and standard deviations are plotted as the blue and red lines, respectively. Each yearly cycle is overlaid as transparent blue or red lines.

and completely ice-full in the winter months. Oceanic flow through Fram Sound initiates sea ice decline in northwestern Jones
Sound typically beginning in April. Sea ice then retreats south across Jones Sound, clearing within the northern fiords first and
then melting back across nearly the entire sound by September. While most of the sound remains ice free in September, Coburg
Island acts as a sea ice bottle neck in our model (and in observations), trapping ice that circulates between the open waters of
Baffin Bay and the confined waters of Jones Sound. During the sea ice maximum, sea ice thickness generally fluctuates between
1-2 m across most of the sound. However, there are isolated pockets of thickness >4 m, which are primarily constrained to
within narrow fiords, erroneously along the periphery of Coburg Island, and erroneously in the open waters of Baffin Bay. We
note that while we only show the 2015 sea ice cycle, these general patterns persist in other modeled years.

To investigate how the yearly sea ice cycle within Jones Sound changes between 2003-2016, we present integrated sea ice
area and thickness in figure 9 across the entire Jones Sound (using the western end of Coburg Island as our eastern limit;





black lines), the western half of Jones Sound (purple lines), and the Eastern half of Jones Sound (green lines). In figure 9,

over these regions of integration, we plot the integrated sea ice maxima (circles) and minima (triangles) and overlay dashed linear trendlines where applicable. In panels b and d, we plot the average 2003-2009 and 2010-2016 yearly cycles of the aforementioned variables and their associated standard deviation, respectively. Beginning with the integrated sea ice area (figure 9a), we observe a temporal decline in minimum sea ice extent of 316 km$^2$/yr in Eastern Jones Sound, mainly associated with enhanced melt of sea ice pinned against Coburg Island and within Eastern Jones Sound (figure 5). When the yearly total

Jones Sound sea ice extent cycles are binned to 2003-2009 and 2010-2016 and averaged (blue and red lines in figure 9b), we further observe that the initiation of sea ice decline is occurring earlier and the onset of sea ice refreezing is occurring later in the 2010-2016 profiles. Furthermore, although we find that while the winter sea ice extent remains stable in our model (Jones Sound becomes completely ice-covered in winter), we observe that the thickness of this winter-ice is declining in time (linear integrated volume trend of -0.384 km$^3$/yr; black-circle line in figure 9c). In fact, the binned-yearly cycles of sea ice volume

reveal that sea ice is generally thinner throughout the entire cycle aside from June and July (figure 9d).

To investigate the impact of tides on sea ice formation and melt within Jones Sound, we execute another simulation that does not include tides and find that tidal flushing through Fram Sound can enhance mean flow velocities by up to 0.75 m/s over that of the non-tidal run in these regions (figure A6). These tidally-enhanced flow velocities through Fram Sound trigger accelerated sea ice meltback in northern Jones Sound between May and July (blue shading in figure A7e-g) while leading to

generally thicker sea ice in the southwest sector of Jones Sound (due to ice advection into the asociated fiords). Similar to figure 9, we integrate sea ice area and volume across Jones Sound and provide the associated time series in figure A8. Zooming into the 2014-2016 cycles, we find that tidal forcing drives a longer ice-free season within Jones Sound, decreasing the summer integrated sea area by up to 6000 km$^2$ (figure A8b-c). Furthermore, tidal forcing also decreases sea ice thickness yearlong due to both enhanced circulation velocities and mixing that entraines heat at depth to the surface (figure A8e-f).

## 3.5 Productivity enhancement

To begin deducing how these temporal changes in the state of Jones Sound sea ice and ocean circulation feedback on photosynthetic biological productivity, we couple our ocean-sea ice model to the MITgcm N-BLING biological productivity module and plot yearly- and depth-integrated profiles of Net Primary Production (NPP; figure 10a), yearly- and depth-integrated profiles of chlorophyll mass (CHL; figure 10b), yearly integrated profiles of Net Community Production (NPP; figure 10c/d), yearly-

integrated profiles of light limitation (figure 10e), and yearly-integrated profiles of iron limitation (figure 10f). Light and iron limitation are computed as the percentage of ocean grid cells per vertical level that experience light or iron limitation, so a value of 100 indicated 100% of ocean grid cells are light or iron limited. NCP represents the difference between gross primary production and total community respiration; that is, when NCP is positive in our model, photosynthetic primary productivity is greater than community respiration and vice versa when negative. In figure 10c/d, we observe that NCP increases in the first

and second vertical ocean level (corresponding to depths of 3.5 m and 10.5 m, respectively, as the tracer point within MITgcm is in the mid-depth of the grid cell). When we zoom into the transition zone between positive and negative NCP, we further observe that the third ocean level (corresponding to a depth of 17.5 m) transitions from pure respiration prior to 2010 to a





source of community production in the fall and summer beyond 2010. To investigate this further, we plot yearly integrated NPP and CHL binned by depth level and observe a strong increase in surface and subsurface NPP after 2010. Prior to 2010,

total time integrated surface (3.5 m depth level) NPP averaged ~0.135 GtC/yr and after 2010, averaged ~0.170 GtC/yr. These productivity enhancements are also elevated at 10.5 m depth (the second ocean depth level), where NPP before and after 2010 average 0.0051 GtC/yr and 0.0075 GtC/yr. We observe that the light limitation time series follows suit, as the mean percentage of summer light limitation of the surface level before and after 2010 is modeled as 48.08% and 43.73%, respectively. Similarly, at 10.5 m depth (the second ocean depth level), the 2005-2010 and 2010-2015 mean summer light limitation is 64.24% and

56.48%, respectively. Interestingly, for the first two ocean depth levels (through 14 m depth), we observe that the percentage of ocean grid cells that are iron limited in the summer increases after 2010; however, for these same levels, winter iron limitation steadily decreases throughout the simulation at a rate of ~0.615%/decade, implying more mixing. For depth levels below 14 m, the yearly cycle of iron limitation decreases linearly at approximately this same rate (yellow-to-red lines in figure 10f). We note that there are no temporal trends evident in the time series of phosphorus and nitrogen limitation and that both fields displayed

decreasing percentages of limitation with depth.

## 4   Discussion

The modeling results presented here highlight the dynamic response of Jones Sound, eastern Canadian Arctic Archipelago (CAA), to changes in the large scale atmospheric and oceanic circulation between 2003-2016. Overall, we find that the primary response of Jones Sound ocean circulation to these past atmospheric and ocean changes is warming of the waters below 200 m

depth. While the magnitude of this warming signal is overestimated in our model due to the Atlantic Water at our Eastern model boundary warming too quickly in winter, we correctly model the depth of the thermocline as well as currents across our model domain, giving us confidence that we correctly simulate the spatial distribution of where these warmer mid-depth waters can circulate to as well as volumetric transport across the domain. In particular, we find that circulation through Lady Ann Strait dominates volumetric transport into and out of Jones Sound, followed by Glacier Strait and Fram Sound. In the latter,

we correctly model the observed ~0.3 Sv of transport (Grivault et al., 2018), although tidal flushing drives high variability. These regions of inflow/outflow drive complex spatial patterns circulation within Jones Sound, with the formation of multiple eddies driving variability in the thermocline depth in the summer. In the winter, under full sea ice coverage, the thermocline shoals under reduced atmospheric forcing. We further find that warm water within Jones Sound is topographically constrained, flowing through Lady Ann Strait and circulating counter-clockwise within the sound, reaching many of the tidewater glaciers

that line the sound's eastern coast (figures 8-A3). In combination with continued observed atmospheric warming (figure 2a), these results underscore the increasing vulnerability of these ice masses to enhanced ocean-induced frontal ablation as CAA waters continue to warm. As enhanced glacial and meltwater discharge can have global impacts via accelerating sea level rise as well as regional impacts via modulating marine productivity through impacts to nutrient supply (Bhatia et al., 2013; Hawkings et al., 2015; Bhatia et al., 2021; Kanna et al., 2018) and fiord-scale estuarine circulation (Lydersen et al., 2014;



**Figure 10.** Bar plots of yearly and depth binned (a) net primary productivity, and (b) chlorophyll. Panels c-d show time series of net community productivity integrated on the top 20 vertical ocean levels (through 140 m depth), where the bold black line in panel-d denotes values for the third vertical ocean level (21 m depth). Panels e and f show the percent of ocean grid cells per each vertical level that are light and iron limited, respectfully.

Straneo and Cenedese, 2015), these simulated changes to coastal Jones Sound waters have implications on both global and local communities.

Furthermore, our modeling results highlight the importance of circulation Fram Sound in triggering sea ice decline in Northern Jones Sound during the summer. Such relationships between tidal forcing and sea ice decline have been studied in other sectors of the Arctic and CAA (e.g., Rotermund et al., 2021; Armitage et al., 2020) but have yet to include Jones Sound due to the relatively small size of the channels that feed this waterway. The complex dependence of Jones Sound sea ice dynamics/thermodynamics on small-scale tidally-forced ocean circulation features (as well as the importance of properly modeling





sea ice change for summertime transportation through these waters) highlights the need for ocean models of this region to both explicitly include tidal forcing and be run at sufficient resolution to resolve the eddies that disperse subsurface heat.

In addition to these sub-annual patterns in simulated sea ice dynamics, we observe long term declines in the Jones Sound integrated summer sea ice extent as well as both the summer and winter sea ice volume. These summertime sea ice extent declines average ∼11%/decade, which are inline with the observed losses over Baffin Bay of 11.7%/decade between 1968-2022 and slightly higher than the 7.1%/decade losses observed across all of Northern Canada's waters over this same time period (Tivy et al., 2011). These losses follow broader patterns of summer Arctic sea ice decline, which are cited to be driven by both natural climate variability (Kinnard et al., 2011; Ding et al., 2017, 2019) as well as human-induced global warming

(Kay et al., 2011; Notz and Stroeve, 2016; Stroeve and Notz, 2018). Here, however, we also show that winter sea ice in Jones Sound is also thinning (figure 9; winter extent does not change in time but winter volume decreases in time), which is likely driven by winter warming of both the subsurface oceanic and surface atmospheric ocean temperatures (figure 2a and figure A5). Lastly, we simulate earlier onset of sea ice decline in the summer and later onset of sea ice refreeze in the fall between 2003-2016. That is, the period of time in which Jones Sound is not completely filled with sea ice is extending in time in our model,

which can have implications on the timing and integrated magnitude of photosynthetic ocean primary productivity.

We observe in our biological modeling results that as the time in which Jones Sound is sea ice free becomes longer, the total integrated oceanic productivity and the depth at which productivity takes place increases. We note that aside from the atmospheric carbon dioxide forcing time series (which largely increases linearly), the boundary conditions in the biological productivity module do not include significant temporal trends and also do not account for nutrient release from enhanced glacial

meltwater/discharge. That is, the response of ocean primary productivity in the results presented here is driven by changes in either local oceanic or atmospheric conditions. Local enhancements to primary productivity have been reported across the Arctic, with the mean annual (March-September) trend of primary productivity increasing ∼50-75 g C/m$^2$/year/decade within the central portion of Jones Sound between 2003-2022 (Frey et al., 2022). While this observed trend includes enhanced productivity due to increased nutrient availability from glacial runoff, the results presented here indicate that changing sea ice and ocean

conditions are also partly responsible for driving these local enhancements to Jones Sound primary productivity. Plausible explanations that could either partly or wholly drive the simulated increase in Jones Sound productivity include: (1) increased availability of light resulting from sea ice decline; (2) increased overturning of the mixed layer from enhanced wind stress as sea ice declines, resulting in greater nutrient upwelling; (3) increased temperature of sub-surface ocean waters can drive enhanced productivity since the carbon-specific photosynthesis rate in N-BLING is temperature-dependent (Galbraith et al.,

2010; Noh et al., 2024). It is expected that a combination of these factors will drive enhanced productivity in our model, and this is evidenced in the light limitation and ocean temperature time series (figure 10e and figure A5, respectively). Specifically, we observe that the pattern of light limitation and ocean temperature through ∼35 m depth mirrors that of the yearly-integrated chlorophyll mass in that beyond 2010, surface and subsurface light limitation and ocean temperature decrease and increase, respectively, driving enhanced productivity. In terms of nutrient limitation, we observe that iron limitation down to ∼14 m depth

increases in the summer and decreases in the winter beyond 2010, highlighting enhanced vertical advection of nutrient-rich waters in the winter that drive productivity blooms once light becomes available. In all, these results highlight that complex



interplay between the atmosphere, ocean, and sea ice will likely continue to drive enhanced productivity in the future under increasing polar-amplified global climate change in Jones Sound.

The results presented here are subject to a high degree of uncertainty that stems from the sparsity of input data used to drive and validate our model as well as processes that are currently unaccounted for. To start, we previously noted that the ANHA12 ocean model output that we used to derive our ocean and sea ice model initial and boundary conditions features large warm biases in winter Atlantic Water (100-300 m), which then propagate throughout our model solution. We selected these boundary conditions because the ANHA12 simulations were high enough resolution to resolve circulation with Jones Sound and the various Straits that feed it and were integrated through the time period of interest. However, circulation within our model is highly dependent on how flow along the oceanic boundary is prescribed. While this is also true for the atmospheric boundary, we are more confident that these fields contain less significant biases than the ocean boundaries. In addition, errors in the bathymetry we use in our model, especially near coastal outlet glaciers, will lead to erroneous paths of warm water circulation. We corrected for bed topography that was too deep near Sverdrup Glacier and too shallow near Grise Fiord; however, it is certain that there are other locations in Jones Sound in which the ocean bathymetry is incorrect and we do not yet have bathymetric observations to apply corrections. For the biological productivity model, recent studies have highlighted the importance of ocean-glacier interactions in driving near-glacier spatial/temporal patterns of productivity within Jones Sound. In particular, subglacial discharge plumes that originate beneath the nutricline can promote vertical advection of nutrients into the euphotic zone while nutrient-rich glacial runoff can feed the upper-ocean; both of these processes have been observed to drive coastal productivity blooms (Achterberg et al., 2018; Bhatia et al., 2021). While we do prescribe runoff as an atmospheric boundary condition in our ocean model, we assume it includes no nutrients and further do not resolve ocean-glacier interactions (including the impact of subglacial discharge plumes). As such, it is likely we do not capture the full extent to which atmospheric and oceanic warming drove change in Jones Sound productivity between 2003-2016 and we flag this as an important next step in this work.

In addition, Jones Sound remains understudied from both a modeling and observational perspective, which limits the amount of publicly available data that can be used as model inputs and validation. On the observational side, it is only since 2019 that recurring observational campaigns have targeted Jones Sound, so repeat oceanographic measurements only exist beyond this date. As such, that ocean models of Jones Sound must be validated based on how well they represent the time-evolving circulation within nearby Baffin Bay and Lancaster Sound, where annual-repeat observations are available since the early 2000's. This is not sufficient, as the modeling results presented here demonstrate that while circulation within Jones Sound is driven by inflow from Baffin Bay and Fram Sound, water masses undergo transformation within Jones Sound and circulation around the sound is sensitive to small-scale bathymetric features. Furthermore, to our knowledge, this is the first publicly available ocean model that was developed and validated with the purpose of studying ocean circulation within Jones Sound, meaning that there exists no other models of sufficient resolution by which we can compare modeling results. As Jones Sound is a critical passageway for liquid transport between the Arctic and Atlantic Oceans and is critical in supporting local communities, we emphasize that future observational campaigns (especially within the critical inflow regions of Lady Ann Strait and Fram



Sound) and modeling studies of Jones Sound should be prioritized so that we can gain a better understanding of the long term impact of global climate change on this region and improve the fidelity of numerical ocean models.

## 5 Conclusions

In this study, we modeled ocean circulation, sea ice dynamics, and biological productivity within Jones Sound, Canadian Arctic
Archipelago, between 2003-2016 with a high-resolution regional configuration of the Massachusetts Institute of Technology general circulation model. Atmospheric forcing was taken from the Arctic System Reanalysis version 2 and ocean boundary conditions were derived from a North Atlantic configuration of the NEMO ocean model (Hu et al., 2019; Gillard et al., 2020). We find that volumetric transport through the three waterways that connect Jones Sound to the Arctic and Atlantic oceans is partitioned as 75%, 14%, and 11% via Lady Ann Strait, Fram Sound, and Glacier Strait, respectively. Surface circulation in the
Summer within Jones Sound is dominated by eddies, whereas winter circulation is quiescent due to sea ice cover. The spatial distribution of summertime eddies varies considerably year-to-year and drives variability in the depth of the thermocline across the sound, impacting the spatial-reach of warm Atlantic Water that sits at depth. This warm water, although topographically constrained, circulates counterclockwise around Jones Sound and expands its spatial reach in the winter when the thermocline shoals. Sea ice dynamics within Jones Sound are sensitive to small-scale circulation features that are generally not resolved
within broad-scale CAA ocean models, such as tidal flushing through Fram Sound which triggers sea ice meltback in the Spring. In addition, we find that wintertime sea ice thickness decreases and the onset of sea ice refreeze in the fall is delayed due to oceanic and atmospheric warming that is simulated in our model. These changes have the impact of lengthening the time and areal extent in which Jones Sound is sea ice free, thus leading to enhanced productivity at all ocean depth levels through 21 meters. While we note that the modeled warming signal in Baffin Bay and Jones Sound is overstated compared to
observations, the results presented here improve our general understanding of circulation into, out of, and within Jones Sound as well as how it impacts sea ice and biological productivity dynamics. These results also emphasize the utility of high-resolution models in simulating complex waterways and underscore the need for sustained oceanographic observations in this region.

*Code and data availability.* All MITgcm parameter files, boundary conditions (including bathymetry), and initial conditions associated with the "high-resolution" Jones Sound ocean-sea ice-biological productivity model, as well as validation data shown in figures, have been
archived in the Dryad Digital Repository (Pelle et al., 2024). These data are available during the peer review process at the following link: http://datadryad.org/stash/share/DHOe1caYiqpIvQm6-fcBuD-r4vL5yoLFztLOvJ2mN3M. MITgcm is also open source and is available for download from mitgcm.org (checkpoint66j). The sea ice and biological productivity (N-BLING) modules are built into MITgcm and are included in its download. Atmospheric forcing used in this study is publicly available via the Arctic System Reanalysis version 2 (ASRv2; Bromwich et al., 2018). Canadian sea ice charts covering the Eastern Arctic are publicly available through the National Snow and Ice Data
Center (Service, 2009). Oceanographic profiles collected aboard the Amundsen Ice Breaker are publicly available through the Polar Data Catalogue (Amu, 2003-2021). Lastly, due to the large amount of model output produced, output is available by request to the corresponding author.



# Appendix A:  Additional Figures

## A1    Additional ocean model figures

**Figure A1.** Vertical ocean temperature profiles (°C) of the Eastern model boundary (in Western Baffin Bay), taken from the ANHA12 NEMO ocean model in January of (a) 2002, (b) 2007, (c) 2012, and (d) 2016. Black shading denotes bedrock.



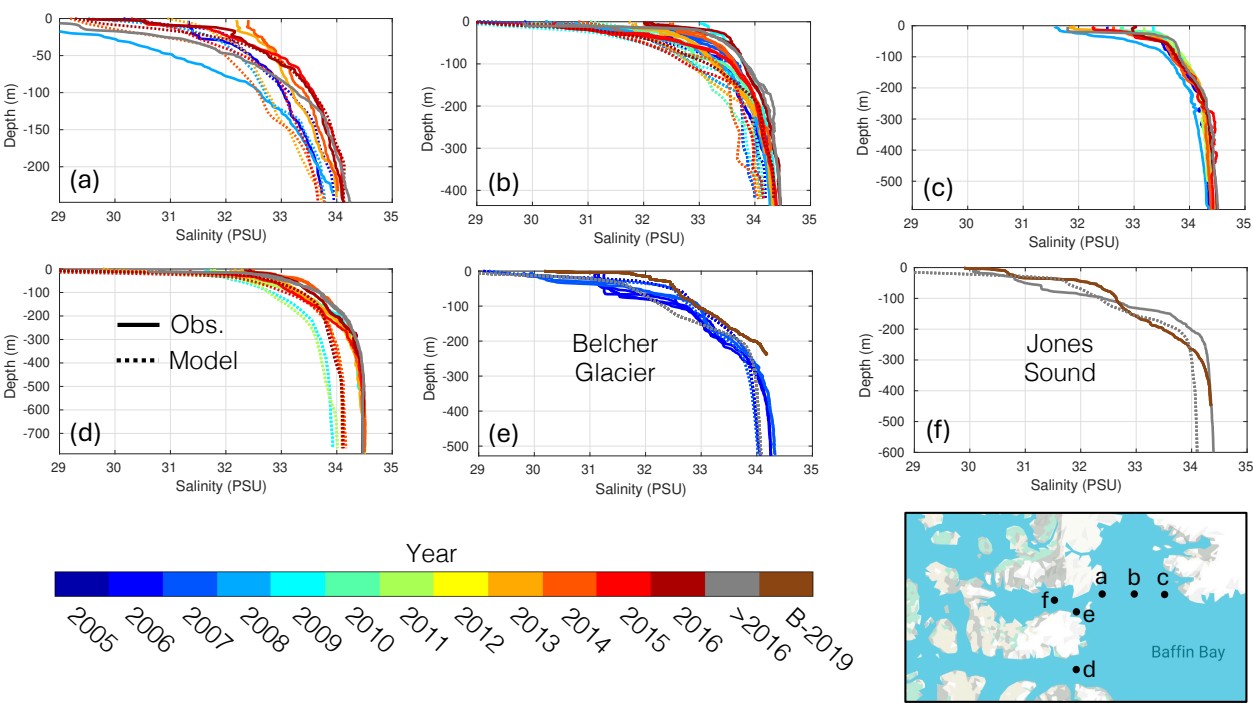

**Figure A2.** Same as in figure 3, but comparing modeled and observed vertical profiles of salinity (units on the Practical Salinity Scale; PSU).



**Figure A3.** Vertical profiles of ocean potential temperature in April of (a-c) 2004, (d-f) 2008, (g-i) 2012, and (j-l) 2016 taken through (a/d/g/j) Sverdrup Bay, (b/e/h/k) Grise Fiord, and (c/f/i/l) the oceanic region adjacent to Belcher Glacier (white, yellow, and red lines in figure 8a). Areas of black denote land and white contours correspond to ocean temperature levels that start at -2°C and increase in 0.5°C increments.



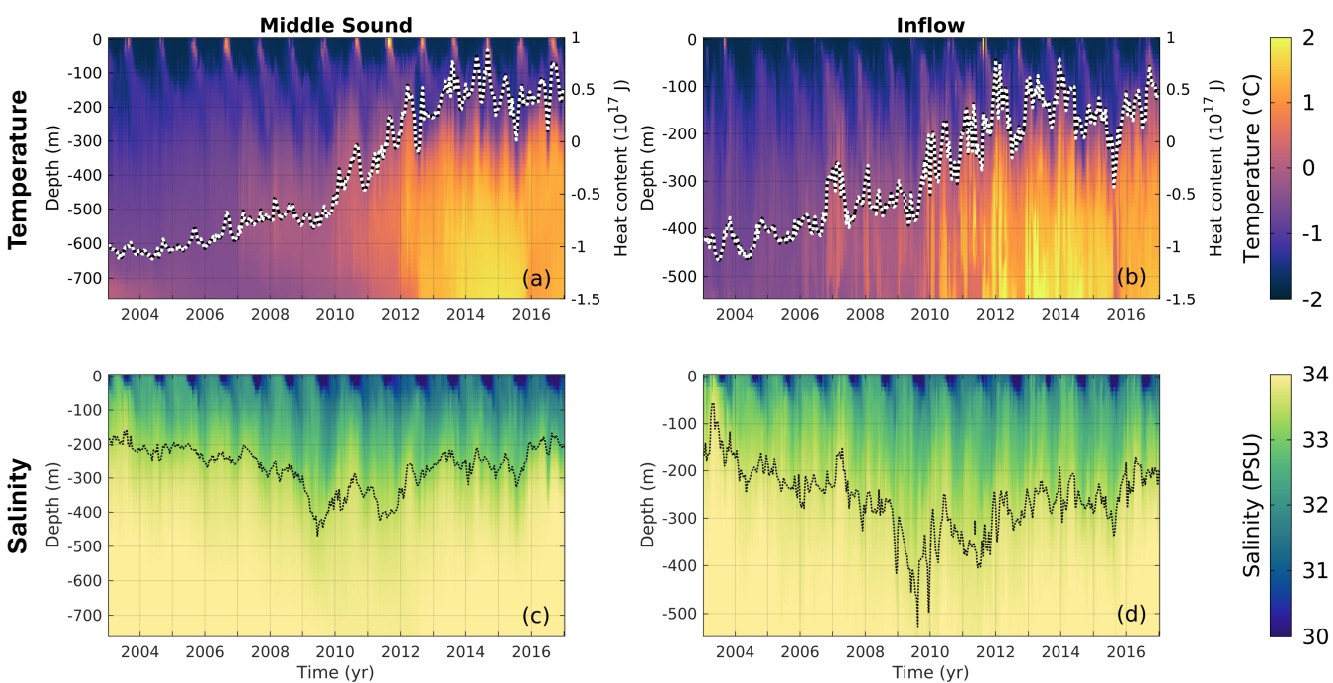

**Figure A4.** Depth-time Hovmöller diagram of (a-b) ocean temperature (°C) and (c-d) salinity (units on the practical salinity scale; PSU) taken at (a/c) the center of Jones Sound (green triangle in figure 8a) and (b/d) the center of Lady Ann Strait (orange circle in figure 8a). In panels a/b, the white-black dashed line is the vertical-integrated ocean heat content (10e17 J) and corresponds to the right y-axis. In panels c/d, the black dotted line is the 33.80 PSU salinity contour.





**Figure A5.** Ocean temperature time series (°C) of the first five ocean depth layers (0-35 m depth) taken within central Jones Sound (green triangle in figure 8a). The black and green dotted lines mark the 2003-2010 mean maximum winter and summer ocean temperatures, respectively.





## A2    Impact of tides on sea ice thickness


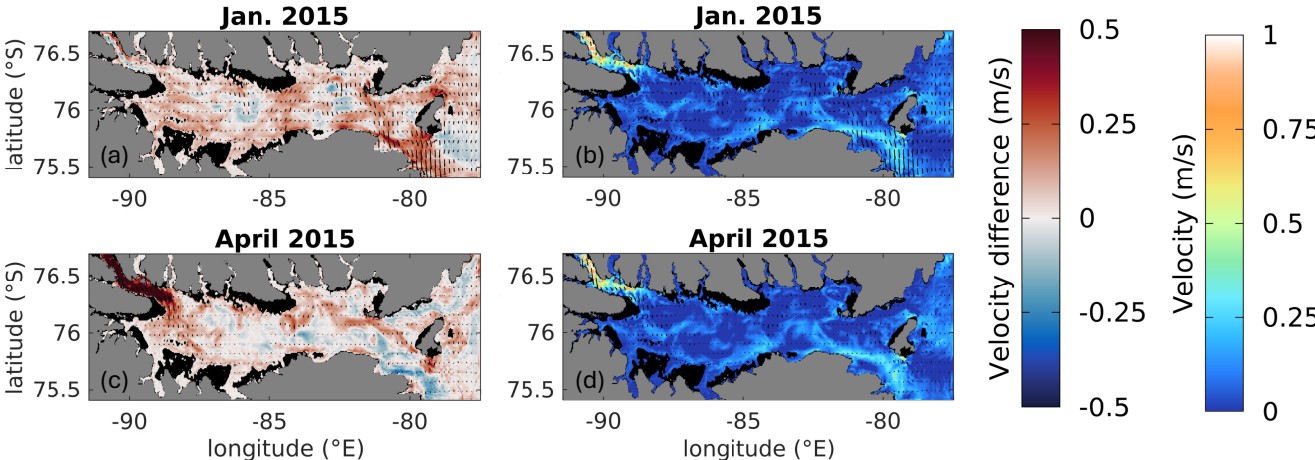

**Figure A6.** Magnitude (b/d) and difference (a/c) in 50 m ocean velocity in (a-b) January 2015 and (c-d) April 2015 between model runs with and without tides (red shading denotes where tidal forcing results in faster ocean velocities). Black shading denotes bathymetry above 50 m and land is shaded in gray. Absolute velocities in panels b/d are taken from the model run with tides.





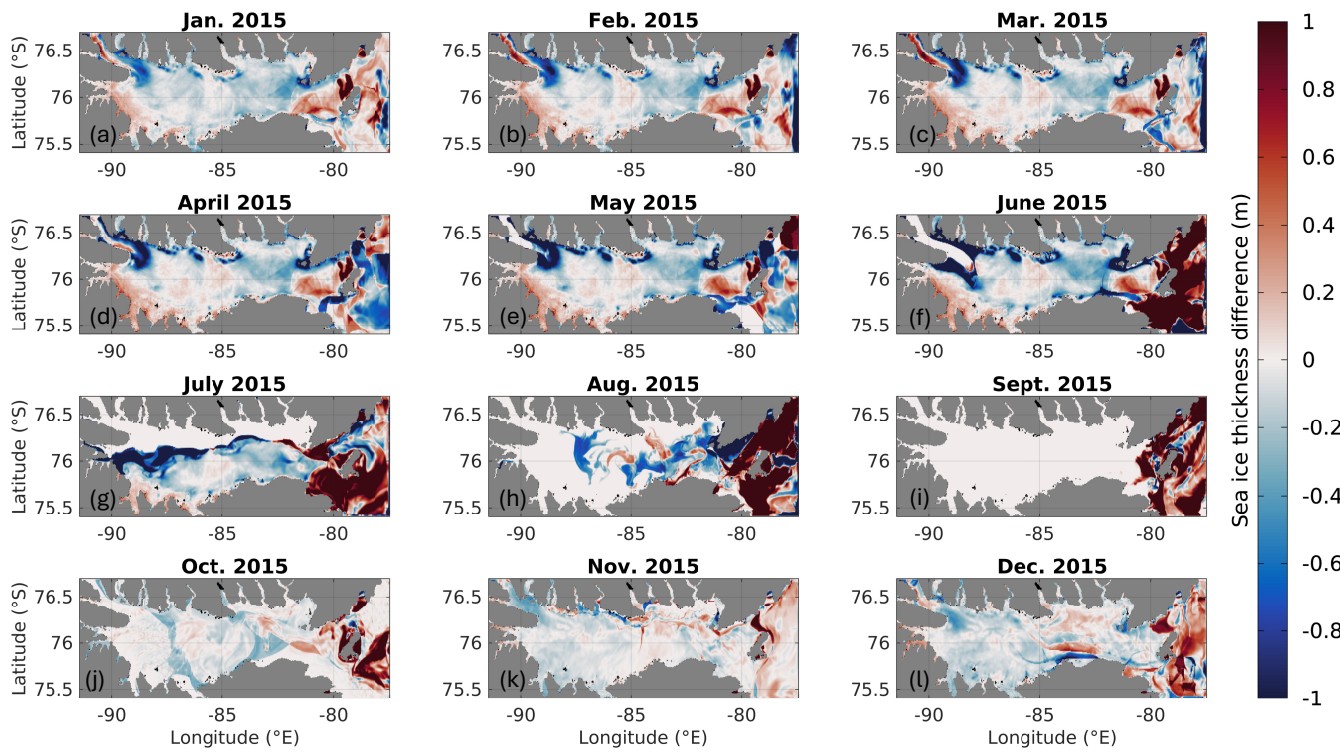

**Figure A7.** Difference in sea ice thickness between model runs with and without tides averaged at each month of the year in 2015 (blue shading denotes where sea ice is thinner in the tidal run).





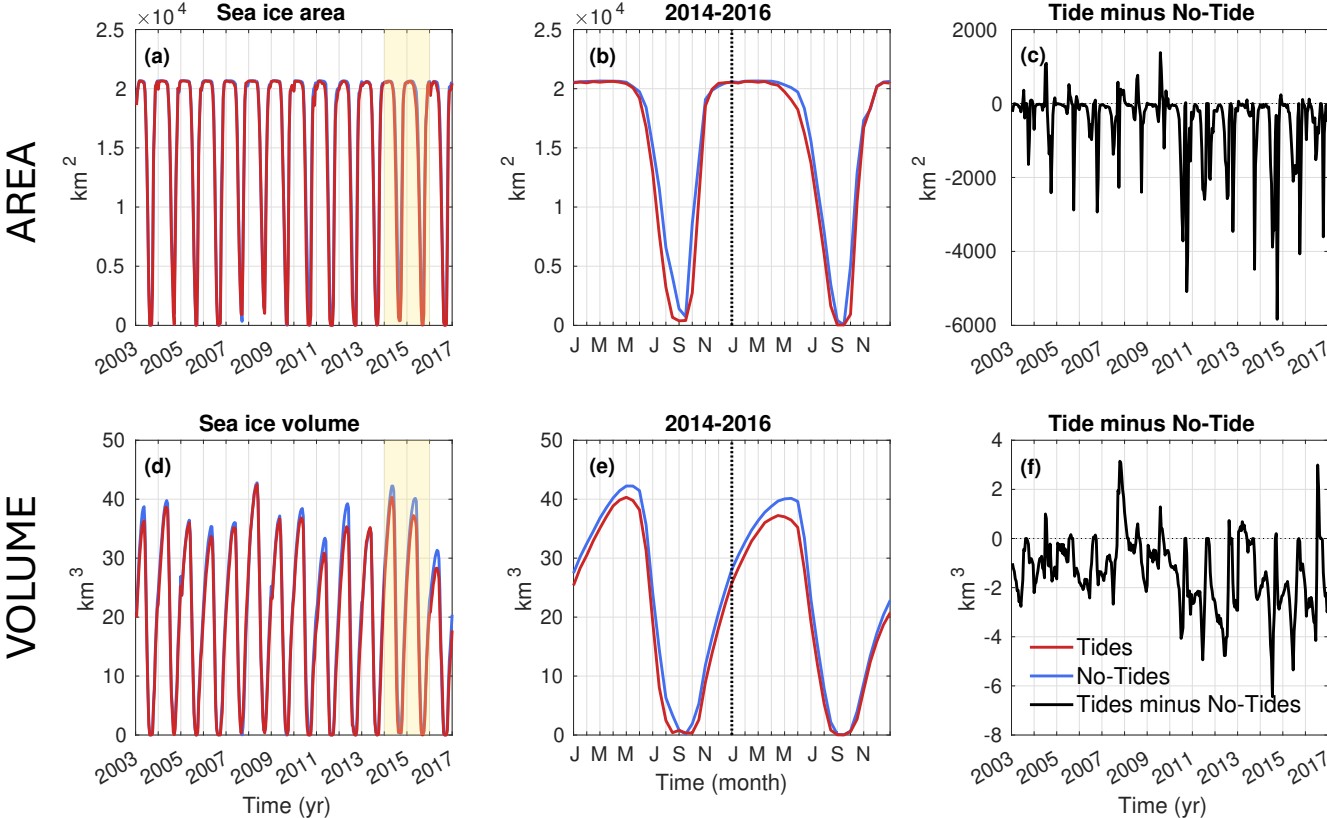

**Figure A8.** Jones Sound integrated sea ice area (a-c) and volume (d-f) timeseries, where red and blue lines represent results from the tidal and non-tidal simulation, respectively. Panels b and e are zoomed into the 2014-2016 sea ice cycle in panels a and d, respectively. In panels c and f, the black line denotes the difference of the tidal and non-tidal time series.



*Author contributions.* TP, JSG, MS, DDB, and KS conceptualized the study with input from all co-authors. PGM and AH provided NEMO ocean model output, updated bathymetry datasets, and domain expertise. MM provided files needed to run N-BLING. TP and KS performed all simulations. TP led manuscript preparation with input from all co-authors. JSG provided TP with supervision and funding.

*Competing interests.* The authors declare no competing interests.

*Acknowledgements.* This project is a result of the international collaborative Exploration of Saline Cryospheric Habitats with Europa Relevance (ESCHER) project with support from NASA grant 80NSSC20K1134. M R Mazloff acknowledges support from NASA Grant 80NSSC24K0243 and NSF Grants OPP-2149501 and OPP-1936222. Oceanographic profiles within the model domain that were presented herein were collected by the Canadian research icebreaker CCGS Amundsen and made available by the Amundsen Science program, which is supported through Université Laval by the Canada Foundation for Innovation.



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
