# Peer review of "Ocean circulation, sea ice, and productivity simulated in Jones Sound, Canadian Arctic Archipelago, between 2003-2016"

_EGUsphere, 2024_

## Author Comment (AC1)

**Response to Reviewer-1**

**Title**: Ocean circulation, sea ice, and productivity simulated in Jones Sound, Canadian Arctic Archipelago, between 2003-2016
**Authors**: T. Pelle, P.G. Myers, A. Hamilton, M. Mazloff, K. Soderlund, L. Beem, D. Blankenship, C. Grima, F. Habbal, M. Skidmore, J.S. Greenbaum
**Journal**: *Ocean Science*

We would first like to thank the editor and two reviewers for their detailed and constructive comments that have vastly improved the manuscript. Please find a point-by-point response to each comment that was raised below. Reviewer comments are presented first and our response is written beneath as indented-bulleted text.

**Reviewer 1**

I am excited to see this type of model developed for investigating circulation, sea ice, and productivity in the Canadian Arctic Archipelago (Jones Sound in this case). I would like to share my own experience with modeling flows through the CAA. Previously, I used a coarse-resolution model to investigate these flows, but I have since recognized the critical need for a high-resolution model to accurately capture detailed current patterns and better understand the circulation and water properties in this region. The authors have done an excellent job in developing this high-resolution model. **It is generally understood that the variations of transports through CAA are mostly controlled by changes in the large-scale circulations, and it would be beneficial to mention this**.

- Thank you! We agree that mentioning this within the text would be helpful, so we added it to the beginning of the introduction where we introduce the CAA (L21-23)

While acknowledging the impressive work, I believe there is still potential for further model improvement. For instance, addressing the warm bias and resolving the model crash issue within the biogeochemical component would enhance its accuracy and reliability. The authors did not investigate the cause of this biogeochemical model crash. While not strictly necessary, providing some insights into this issue would be valuable for the scientific community.

- The Atlantic Water warm bias in our model is unfortunately sourced from the ANHA12 NEMO run which we took our boundary conditions from (rather than from an issue with our MITgcm model). Addressing this would require us to redevelop the model from scratch, which unfortunately is too large of a task. However, we purposefully limited analysis of the model outputs to circulation, volume transport, sea ice, and near-surface biological productivity, which should be minimally impacted by the warm bias. However, we agree that fixing this would improve the reliability of our model and flag this as an important step in future work. Furthermore, we highlight that the boundary conditions are the source of the warm bias in both the Results Section (L139-142, L161, ) and

Discussion Section (L379-384).

- We agree that investigating the reason for the crash in the biogeochemical model would be valuable to the scientific community and for those using N-BLING on other polar regions. We dug into this issue and found the exact location and cause of the model crash. We included a new paragraph in the discussion section that fully describes the reason for the crash and ways to avoid this in future studies (L369-3798).

Line 65: "?" is needed at the end of the (2)

- Done.

Lines 132-33: 2m air temperature is used in driving the model, but why 10 m air temperature is used here?

- Thank you for catching this, we corrected this here and in other places in the manuscript.

---

## Author Comment (AC2)

**Response to Reviewer-2**

**Title**: Ocean circulation, sea ice, and productivity simulated in Jones Sound, Canadian Arctic Archipelago, between 2003-2016
**Authors**: T. Pelle, P.G. Myers, A. Hamilton, M. Mazloff, K. Soderlund, L. Beem, D. Blankenship, C. Grima, F. Habbal, M. Skidmore, J.S. Greenbaum
**Journal**: *Ocean Science*

We would first like to thank the editor and two reviewers for their detailed and constructive comments that have vastly improved the manuscript. Please find a point-by-point response to each comment that was raised below. Reviewer comments are presented first and our response is written beneath as indented-bulleted text.

**Reviewer 2**

The model simulation stops in 2016. Would it be possible to extend the simulation until recent years?

- The end date of the simulation was constrained to 2016 due to the availability of external forcing from the Arctic Systems Reanalysis version 2, which was chosen because of its high resolution and calibration for Arctic regions. We are also constrained by the runtime of the ANHA NEMO model run that we used to derive our coarse-model boundary and initial conditions. Lastly, we were also constrained a bit by the availability of computational resources for this project, so we unfortunately do not have the capacity to run the models out closer to the present-day.

It is a bit confusing that you use two MITgcm setups here, plus the NEMO simulation used for the boundary conditions. Why is the low resolution MITgcm simulation needed? Can't you apply to boundary conditions from NEMO directly to the high resolution MITgcm setup?

- We could technically apply the NEMO boundary conditions directly to the high resolution Jones Sound model, but the resolution of the NEMO model (1/12 degree) is 10x coarser than that of the MITgcm model (1/120 degree). It is generally best practice to refine the resolution of nested models by no more than 5x, which results in needing an intermediate model (the "coarse" MITgcm model) to "safely" refine to the 1/120 degree resolution. In addition, the width of Cardigan Strait is close to the NEMO grid resolution, meaning that flow prescribed at the boundary near Cardigan Strait would be governed by a single grid cell, which would likely drive errors in simulated flow through Fram Sound. For these reasons, we found it necessary to first use a coarse resolution MITgcm model to refine flow through the region and then use that model as boundary conditions on our high resolution Jones Sound model.

My major point concerns the volume transports described on page 11. I am quite confused about that. The net flow through Jones Sound should amount to zero, otherwise there would be an accumulation/volume loss. So why is the net volume transport (Figure 6a/b) not zero? Is there a major contribution from runoff? I also find the sentence in line 192-194 confusing, as the sum of the absolute value of inflow and outflow is not very meaningful. Please clarify. The sentence in the abstract (lines 7-9) should also be checked. Is this related to the way you apply boundary conditions (no flow at boundaries) in the model?

- Thank you for catching this, we were using the output velocity fields to compute transports but realized that this method does not account for the nonlinear surface. Because of this, we re-ran both the tidal and non-tidal models to output the actual mass flux fields and recomputed the correct volumetric transports, which are now plotted in figure-6. In panels a/b, we added the magnitude of transport through each sound as a thin dotted line, as well as a black-dashed line that shows the net volume transport (which now sums very close to zero for both models).

The discussion section reads in parts like a conclusion. It could be made clearer what is actually discussed here. Maybe adding subtitles would help.

- We have added subsections to the discussion section to help orient the reader. We also modified the discussion text to be more succinct.

Line 12: the 11% reduction, is this for the time period 2003-2016? Maybe add the time period here.

- Done

Line 28: add a reference for the 0.3 Sv for Jones Sound

- Done.

Lines 64-66: The points are a bit out of context, since you already stated the aims of the study in Lines 57-60.

- Thank you for catching this, that was left over text that was supposed to be commented out of the *.tex file. We have removed it.

Line 168: remove "and 2019"

- We modified this sentence and fixed this issue.

Line 217: why do you take 150 m? Instead of taking the upper 150 m, you could take T=0°C as a threshold and plot the mean circulation in the water layer with T<0°C.

- We tried to implement this but there are several regions where the entire water column has temperature below 0°C, in particular within Fram Sound, coastal sectors of Jones Sound, and within Glacier Strait (regions where the bathymetry is quite shallow

compared to central Jones Sound and Lady Ann Strait). This causes us to lose visuals of circulation in these waterways, so we have decided to retain the 150 m threshold.

Lines 273 and 275: should this be ice area and not extent? In the figure ice area is shown. It would also make sense to define "ice area" somewhere in the text.
- That is correct - we added a definition of sea ice area in this paragraph and changed "extent" to "area" elsewhere in the text.

Line 294: abbreviation should be NCP
- Done.

Line 337: --> importance of circulation in Fram Sound
- Done.

Line 346: --> ...declines on average
- Done.

Figure 3: The plots showing observed and modeled profiles is not very clear, there are too many lines on top of each other. Maybe there is a better way to visualize the comparison.
- We have updated both figures (figure-3 in the main manuscript and figure A2 in the appendix) to separate the observed and modeled profiles so that there are not as many lines overlapping. Where applicable, we also reduced the number of profiles to 1 per year so that there are not overlapping lines of the same color. Lastly, we updated the map in this figure so that the profile locations are easier to discern.

Figures 4 and 5 could be plotted in the same style, that would look nicer.
- We have updated the sea ice colorbar and land color of figure-5 to match that of figure-4 so the figures look more similar. Unfortunately, the Canadian Ice Service only outputs the historic sea ice concentration fields as polygons, so they will not plot and have the same style as the MITgcm output (i.e., the polygons will always have hard cutoffs while the model data will have smoother cutoffs).

Figure 7: the arrows are very hard to see. Is it possible to increase the panel size (decrease the space between panels)?
- We increased the panel size and also reduced the number of arrows in panels a-f so that their size is increased a bit.

Figure 9 caption: c-d is volume, and not extent, right?
- That is correct, thank you for catching that!